# Deep mutational scans for ACE2 binding, RBD expression, and antibody escape in the SARS-CoV-2 Omicron BA.1 and BA.2 receptor-binding domains

Tyler N. Starr[1,2‡]*, Allison J. Greaney[1,3,4‡], Cameron M. Stewart[5], Alexandra C. Walls[5], William W. Hannon[1,6], David Veesler[5,7], Jesse D. Bloom [1,3,7]*

1 Basic Sciences Division, Fred Hutchinson Cancer Research Center, Seattle, Washington, United States of America, 2 Department of Biochemistry, University of Utah, Salt Lake City, Utah, United States of America, 3 Department of Genome Sciences, University of Washington, Seattle, Washington, United States of America, 4 Medical Scientist Training Program, University of Washington, Seattle, Washington, United States of America, 5 Department of Biochemistry, University of Washington, Seattle, Washington, United States of America, 6 Molecular and Cellular Biology Graduate Program, University of Washington, Seattle, Washington, United States of America, 7 Howard Hughes Medical Institute, Seattle, Washington, United States of America

‡ These authors share first authorship on this work.
* tstarr@fredhutch.org (TNS); jbloom@fredhutch.org (JDB)

**Data Availability Statement:** Site saturation mutagenesis libraries are available from Addgene (accession # 1000000187 and 1000000188 for

## Abstract

SARS-CoV-2 continues to acquire mutations in the spike receptor-binding domain (RBD) that impact ACE2 receptor binding, folding stability, and antibody recognition. Deep mutational scanning prospectively characterizes the impacts of mutations on these biochemical properties, enabling rapid assessment of new mutations seen during viral surveillance. However, the effects of mutations can change as the virus evolves, requiring updated deep mutational scans. We determined the impacts of all single amino acid mutations in the Omicron BA.1 and BA.2 RBDs on ACE2-binding affinity, RBD folding, and escape from binding by the LY-CoV1404 (bebtelovimab) monoclonal antibody. The effects of some mutations in Omicron RBDs differ from those measured in the ancestral Wuhan-Hu-1 background. These epistatic shifts largely resemble those previously seen in the Alpha variant due to the convergent epistatically modifying N501Y substitution. However, Omicron variants show additional lineage-specific shifts, including examples of the epistatic phenomenon of entrenchment that causes the Q498R and N501Y substitutions present in Omicron to be more favorable in that background than in earlier viral strains. In contrast, the Omicron substitution Q493R exhibits no sign of entrenchment, with the derived state, R493, being as unfavorable for ACE2 binding in Omicron RBDs as in Wuhan-Hu-1. Likely for this reason, the R493Q reversion has occurred in Omicron sub-variants including BA.4/BA.5 and BA.2.75, where the affinity buffer from R493Q reversion may potentiate concurrent antigenic change. Consistent with prior studies, we find that Omicron RBDs have reduced expression, and identify candidate stabilizing mutations that ameliorate this deficit. Last, our maps highlight a broadening of the sites of escape from LY-CoV1404 antibody binding in BA.1 and BA.2 compared to the ancestral Wuhan-Hu-1 background. These BA.1 and BA.2

BA.1 and BA.2 libraries, 184408 and 184409 for respective parental constructs). Raw sequencing data are on the NCBI SRA under BioProject PRJNA770094, BioSamples SAMN30603816 (PacBio sequencing), SAMN30603946 (Illumina barcode sequencing for ACE2 binding and expression experiments), and SAMN30603977 (Illumina barcode sequencing for antibody escape mapping). All code and data at various stages of processing is available at https://github.com/jbloomlab/SARS-CoV-2-RBD_DMS_Omicron and https://github.com/jbloomlab/SARS-CoV-2-RBD_Omicron_MAP_LY-CoV1404. Outlines of the analytical pipelines and links to descriptive notebooks for each analytical step are available at https://github.com/jbloomlab/SARS-CoV-2-RBD_DMS_Omicron/blob/main/results/summary/summary.md and https://github.com/jbloomlab/SARS-CoV-2-RBD_Omicron_MAP_LY-CoV1404/blob/main/results/summary/summary.md. Final mutant deep mutational scanning measurements are available in S1-S3 Data, and interactive visualizations of key data are available at: https://jbloomlab.github.io/SARS-CoV-2-RBD_DMS_Omicron/.

**Funding:** This project has been funded in part with federal funds from the NIAID/NIH under contract numbers 75N93022C00036 to D.V. and 75N93021C00015 to J.D.B. This work was supported by grants from the NIAID/NIH (K99AI166250 to T.N.S., T32AI083203 to A.J.G., DP1AI158186 to D.V. and R01AI141707 to J.D.B.), NIGMS/NIH (R01GM120553 to D.V.), the Gates Foundation (OPP1156262 to D.V. and INV-004949 to J.D.B.), a Pew Biomedical Scholars Award (D.V.), an Investigators in the Pathogenesis of Infectious Disease Award (D.V.), and Fast Grants (D.V.). T.N.S. was a Howard Hughes Medical Institute Fellow of the Damon Runyon Cancer Research Foundation (DRG-2381-19). D.V. and J.D.B. are Investigators of the Howard Hughes Medical Institute. The funders had no role in study design, data collection and analysis, decision to publish, or preparation of the manuscript.

**Competing interests:** I have read the journal's policy and the authors of this manuscript have the following competing interests: J.D.B. has consulted for Moderna and Merck on viral evolution and epidemiology. T.N.S. and J.D.B. consult for Apriori Bio on deep mutational scanning. T.N.S., A.J.G., and J.D.B. receive a share of intellectual property revenue as inventors on Fred Hutchinson Cancer Center–optioned technology and patents related to deep mutational scanning of viral proteins and stabilization of SARS-CoV-2 RBDs.

deep mutational scanning datasets identify shifts in the RBD mutational landscape and inform ongoing efforts in viral surveillance.

## Author summary

SARS-CoV-2 evolves in part through mutations in its spike receptor-binding domain. As these mutations accumulate in evolved variants, they shape the future evolutionary potential of the virus through the phenomenon of epistasis. We characterized the functional impacts of mutations in the Omicron BA.1 and BA.2 receptor-binding domains on ACE2 receptor binding, protein folding, and recognition by the clinical LY-CoV1404 antibody. We then compared the measurements to prior data for earlier variants. These comparisons identify patterns of epistasis that may alter future patterns of Omicron evolution, such as turnover in the availability of specific affinity-enhancing mutations and an expansion in the number of paths of antibody escape from a key monoclonal antibody used for therapeutic treatment of COVID-19. This work informs continued efforts in viral surveillance and forecasting.

## Introduction

The SARS-CoV-2 spike receptor-binding domain (RBD) continues to evolve via substitutions that balance ACE2 receptor binding with escape from neutralizing antibodies. Deep mutational scanning has provided experimental maps for evaluating how RBD mutations impact these key properties [1–5]. These maps enable the immediate evaluation of the possible impacts of new mutations observed during viral surveillance and inform efforts in viral forecasting [6–8].

However, the functional impacts of amino acid mutations are not constant over time. Due to the phenomenon of epistasis, substitutions that accrue as a protein evolves often modify the functional effects of other mutations [9]. Due to these epistatic shifts, deep mutational scans carried out in new RBD variant backgrounds are necessary to continue interpreting and modeling viral evolution [10].

Omicron marks the largest evolutionary jump in SARS-CoV-2 RBD sequence evolution seen to date. The BA.1 and BA.2 Omicron variants have 15 and 16 amino acid substitutions in the RBD alone (Fig 1A) [11,12], many of which impact ACE2 binding or antibody recognition [5,13–17]. Though it is now clear that epistatic interactions among substitutions shaped Omicron emergence [10,16,18–21], it is unclear how the substitutions in Omicron have reshaped the impacts of all the other possible mutations that form the substrate for future viral evolution.

## Results

### Deep mutational scans of the Omicron BA.1 and BA.2 RBDs

We performed deep mutational scanning experiments to characterize the functional impacts of all single mutations in the Omicron BA.1 and BA.2 RBDs on ACE2-binding affinity and RBD expression, a proxy for protein stability [22]. We constructed duplicate site-saturation mutagenesis libraries in the Omicron BA.1 and BA.2 RBD backgrounds (S1 Fig) and cloned the mutant libraries into a yeast-surface display platform [23] alongside a previously cloned

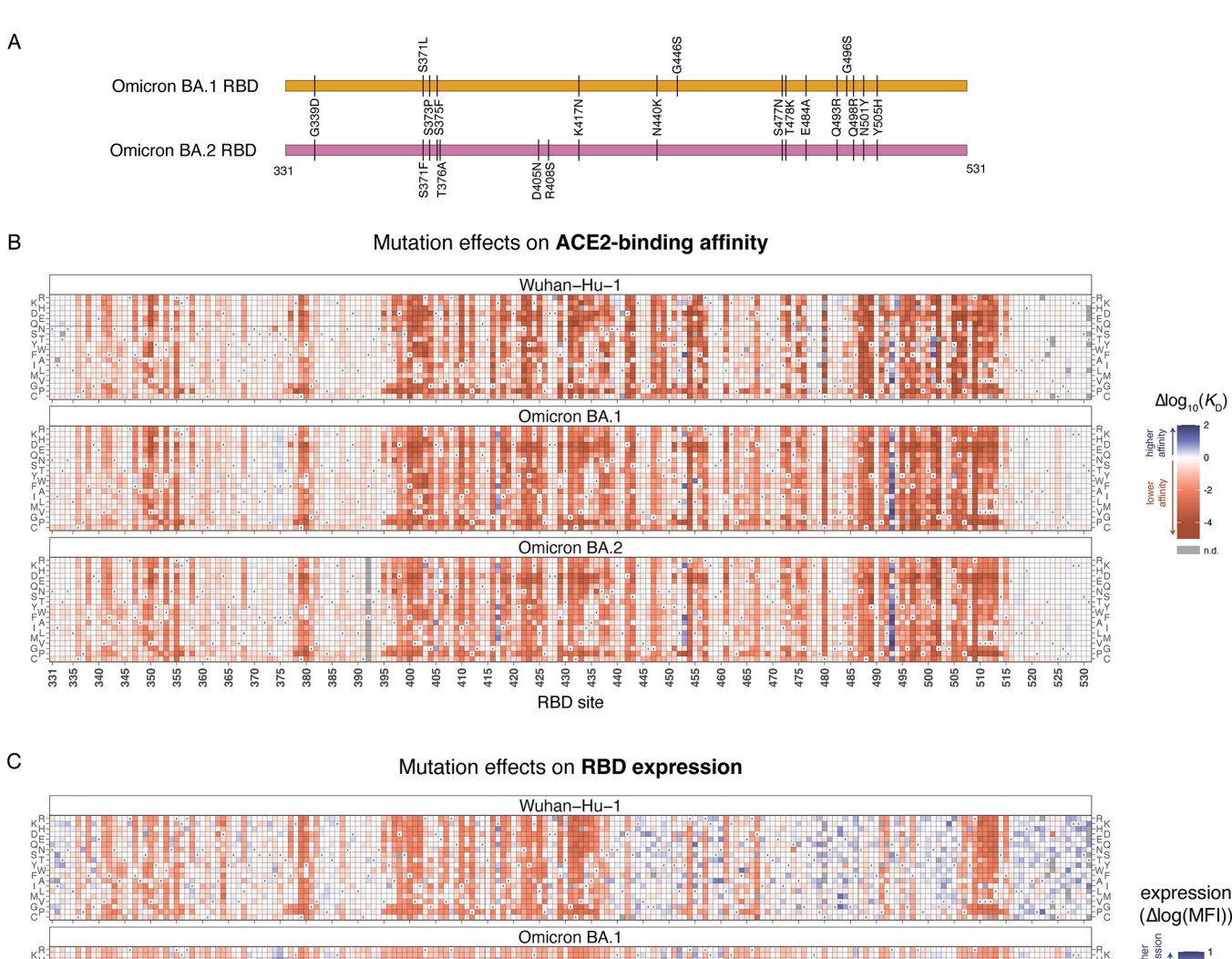

**Fig 1. Effects of mutations in Omicron BA.1 and BA.2 receptor-binding domains on ACE2-binding and RBD expression. (A)** Diagram of the RBD substitutions that distinguish Omicron BA.1 and BA.2 from Wuhan-Hu-1. We use Wuhan-Hu-1 spike numbering throughout. **(B, C)** Heatmaps illustrating the impacts of all single mutations on ACE2-binding affinity (B) and RBD surface expression (C), as determined from FACS-seq assays on yeast-displayed RBD mutant libraries. See S1–S3 Figs. for experimental details. Individual measurements are reported in S1 and S2 Data, and an interactive version of these heatmaps is available at https://jbloomlab.github.io/SARS-CoV-2-RBD_DMS_Omicron/RBD-heatmaps/.

library in the ancestral Wuhan-Hu-1 background [10]. To determine the impacts of RBD mutations on ACE2-binding affinity, we induced surface expression of the pooled yeast-displayed mutant libraries and incubated the libraries across a concentration gradient of monomeric human ACE2. We previously found that monomeric ACE2 better resolves small and

moderate positive or negative mutational effects on binding compared to dimeric ACE2 [10]. We used fluorescence-activated cell sorting (FACS) followed by deep sequencing to quantify the strength of ACE2 binding of each library mutant at each ACE2 concentration, enabling the calculation of a dissociation constant ($K_D$) from binding curves for each mutant in the library (S2 Fig and S1 Data) [24]. An analogous FACS-seq process was used to quantify RBD surface expression based on fluorescence detection of a C-terminal epitope tag on the yeast-displayed RBD (S3 Fig and S2 Data).

Heatmaps illustrating the effects of each mutation on ACE2-binding affinity and RBD expression are shown in Fig 1B and 1C, and as an interactive figure at https://jbloomlab. github.io/SARS-CoV-2-RBD_DMS_Omicron/RBD-heatmaps/ (which also includes previously published deep mutational scanning measurements for the Alpha, Beta, Delta, and Eta variant RBDs [10]). Like prior deep mutational scans in other RBD backgrounds [1,10], we find that the Omicron BA.1 and BA.2 RBDs are highly tolerant to mutation and can sample mutations that increase affinity for ACE2 (Fig 1B). In the case of Omicron, some of these affinity-enhancing mutations are reversions or secondary changes at sites that mutated during Omicron's emergence (e.g., mutations of residues N417 or R493).

We found that Omicron BA.1 and BA.2 RBDs have reduced yeast surface expression levels relative to Wuhan-Hu-1 (S2 Data and S3C and S3D Fig), consistent with biochemical studies demonstrating that the Omicron RBD is less thermodynamically stable than Wuhan-Hu-1 [16,25,26]. Our maps identify candidate stabilizing mutations in the Omicron RBD core (e.g., positions 358, 363, 365, 392) and surrounding the loop containing Omicron substitutions at sites 371, 373, and 375 (e.g., positions 369, 374, 376; Fig 1C). We previously showed that space-filling core mutations such as I358F, Y365W, and F392W greatly increase stability and soluble yield of mammalian-expressed Wuhan-Hu-1 RBD [1] and correspondingly increase the yield and stability of an RBD-based nanoparticle vaccine without altering antigenicity [27]. Notably, the "rpk9" combination of stabilizing mutations (Y365F, F392W, V395I) described for the Wuhan-Hu-1 RBD [27] also enhanced yeast-surface expression of the Omicron BA.1 RBD (S3D Fig).

## Epistatic shifts in the Omicron mutational landscape

We next determined how the functional impacts of individual amino acid mutations differ between Omicron backgrounds and the ancestral Wuhan-Hu-1 strain due to epistasis. We computed an "epistatic shift" metric that identifies positions in the Omicron BA.1 or BA.2 RBDs with large changes in the effects of mutations compared to Wuhan-Hu-1 (Fig 2A, interactive plot available at https://jbloomlab.github.io/SARS-CoV-2-RBD_DMS_Omicron/ epistatic-shifts/) [10]. This epistatic shift metric scales from 0 for a site where the distribution of 20 amino acid affinities is identical between backgrounds to 1 for a site where the distributions are entirely dissimilar (see S4 Fig for representative mutational-level shifts underlying different sitewise epistatic shift values). The pattern of epistatic shifts in the BA.1 and BA.2 RBDs closely mirrors that previously described in the Alpha variant (which contains N501Y alone) (Fig 2A and 2B) and the Beta variant (which contains N501Y) (S4 Fig) [10], suggesting that the shared N501Y substitution remains a dominant determinant of the mutational landscape. However, there are several variant-specific epistatic shifts evident in the Omicron data. Sites 439, 453, and 455 did not change between Wuhan-Hu-1 and Omicron, but these positions exhibit epistatic shifts that alter the availability of affinity-enhancing mutations (Figs 1B and S5). For example, the N439K mutation that enhances the affinity of the Wuhan-Hu-1 RBD for ACE2 and occurred convergently in several early SARS-CoV-2 variants [28] is affinity-decreasing in Omicron RBDs. In contrast, the mutation L455W which reduces ACE2

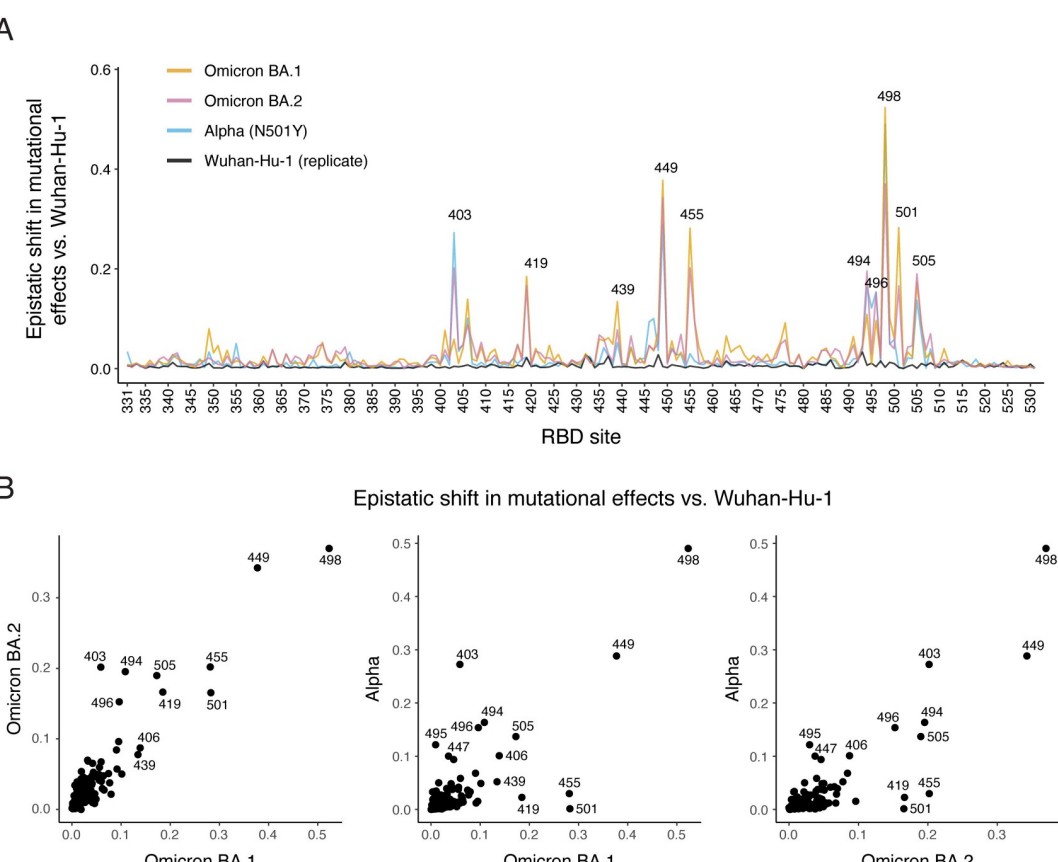

**Fig 2. Epistatic shifts in mutational effects on ACE2 binding. (A)** Epistatic shift in the effects of mutations on ACE2 binding at each RBD position compared to Wuhan-Hu-1. Interactive plot is available at https://jbloomlab.github.io/SARS-CoV-2-RBD_DMS_Omicron/epistatic-shifts/. Alpha and Wuhan-Hu-1 (replicate) datasets are from previously described deep mutational scanning [10]. Epistatic shifts of all variants are shown in S4 Fig. Epistatic shift represents the Jensen-Shannon divergence in the Boltzmann-weighted affinities for all amino acids at each site. See S5 Fig for patterns of mutation-level shifts at relevant sites. **(B)** Scatterplots of epistatic shifts versus Wuhan-Hu-1 at each RBD position between Omicron BA.1, BA.2, and Alpha backgrounds. Points that fall off of the 1:1 line indicate sites where mutational effects diverge from Wuhan-Hu-1 in one background more than the other.

affinity in Wuhan-Hu-1 is affinity-enhancing in Omicron RBDs. While L455W is accessible via single-nucleotide mutation from the TTG codon present in Wuhan-Hu-1 and Omicron, mutations to site 455 have not yet been important in SARS-CoV-2 variant evolution (but see [29]). This may be because all such mutations previously decreased ACE2 affinity. But site 455 is variable across the broader evolution of SARS-related coronaviruses, e.g., Y455 in SARS-CoV-1 and W455 in RsSHC014 (SARS-CoV-2 numbering), and the epistatic shift that makes some mutations at this site favorable for ACE2 binding in Omicron suggests it could become relevant for future SARS-CoV-2 evolution.

Other notable epistatic shifts between Wuhan-Hu-1 and Omicron occur at positions that changed in BA.1 and/or BA.2 (Fig 1A). Several substitutions in Omicron such as N501Y and Q498R exhibit epistatic entrenchment, where the derived state is more favorable for ACE2 binding in the fully evolved Omicron background than in Wuhan-Hu-1 (Figs 3A and S6). For example, the N501Y mutation in the Wuhan-Hu-1 background has a positive effect on ACE2 binding affinity of $\Delta\log_{10}K_D = 1.07$ (that is, the mutation enhances affinity 11.7-fold). If there were no epistasis, then the Y501N reversion in Omicron RBDs would have an opposite-sign effect of $\Delta\log_{10}K_D = -1.07$. However, the measured effect of Y501N is $\Delta\log_{10}K_D = -2.46$ in

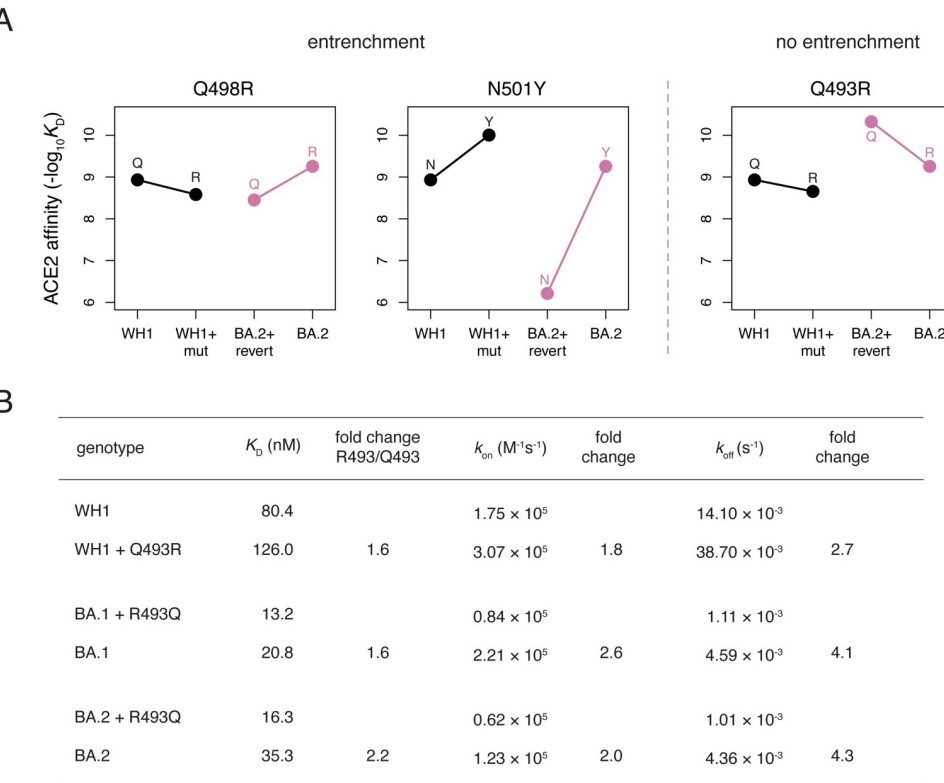

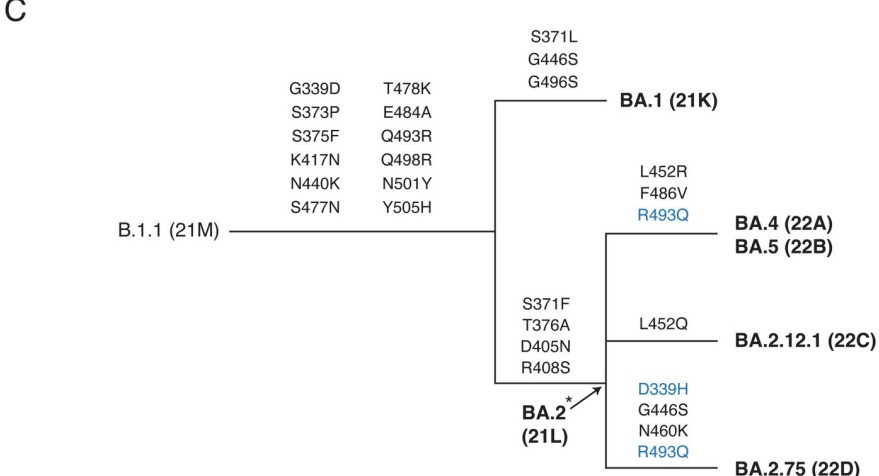

**Fig 3. Entrenchment (or lack of entrenchment) of Omicron substitutions. (A)** Patterns of entrenchment (or lack thereof) of Omicron substitutions Q498R, N501Y, and Q493R. Each plot shows the effect of the labeled mutation ("mut") as measured in Wuhan-Hu-1 (WH1, black) or its reversion ("revert") in Omicron BA.2 (pink). A difference in the slope of the lines connecting the ancestral (left) to derived (right) state affinities illustrates epistasis. Entrenchment is evident in mutations where the slope of the pink line is greater (more positive or less negative) than the slope of the black line and arises from favorable epistasis between the entrenched substitution and one or more other co-occurring substitutions. See S6 Fig for entrenchment patterns at all substituted Omicron BA.1 and BA.2 positions. **(B)** Measurement of binding kinetics of Q493R mutants via biolayer interferometry (BLI). Kinetic parameters and fold-change of R493 compared to Q493 value are given. See S7 Fig for raw BLI sensorgram traces and parameters from a second biological replicate. **(C)** Schematic cladogram illustrating the relationship of major Omicron sub-variants and history of RBD substitutions. Secondary substitutions and reversions of basal Omicron substitutions are marked in blue. Asterisk indicates that the representation of BA.2 being a direct ancestor of all shown descendant lineages is approximate/uncertain.

BA.1 and $\Delta\log_{10}K_D = -3.04$ in BA.2 (that is, the reversion reduces affinity 288- and 1096-fold). Entrenchment is a common phenomenon in protein evolution that arises when subsequent mutations are contingent on earlier ones [30,31], as has been suggested for Omicron substitutions [18,19]. These patterns of entrenchment suggest that reversion of substitutions like N501Y is unlikely to occur in subsequent Omicron evolution.

In contrast, the Q493R substitution does not show any evidence of entrenchment (Fig 3A and 3B). Instead of becoming more tolerable in combination with the other Omicron mutations, as would occur for entrenchment, R493 is more unfavorable for ACE2 binding in the BA.1 and BA.2 backgrounds in the yeast-display assay (Fig 3A). A similar phenomenon was seen in binding kinetics determined via biolayer interferometry (BLI) with purified mammalian-expressed proteins (Figs 3B and S7): the R493 state causes a greater increase in the off rate ($k_{off}$) relative to Q493 in Omicron backgrounds compared to Wuhan-Hu-1 (Fig 3B). R493 causes a concurrent increase in $k_{on}$ in Omicron such that the fold-decrease in $K_D$ caused by Q493R is similar in Wuhan-Hu-1 and Omicron backgrounds, reflecting a lack of entrenchment of this substitution. The difference in $K_D$ fold-change seen between BLI and yeast-display assays may reflect a bias against elevated $k_{off}$ in the yeast assay, where measurement is performed following post-equilibrium wash steps that may dissociate bound ligand whereas BLI measures binding kinetics in real-time.

In contrast to entrenchment which disfavors reversion over time, the lack of entrenchment of the affinity-decreasing Q493R substitution may promote its subsequent reversion. Indeed, the R493Q reversion has occurred multiple times in recent Omicron evolution, including the highly successful Omicron sub-variants BA.4, BA.5, and BA.2.75 (Fig 3C) [12]. The gain in ACE2-binding affinity conferred by the R493Q reversion likely enables additional mutations in these variants like F486V, which reduces affinity for ACE2 but enables further immune escape [17,32–34]. A similar pattern of reversion of a recent destabilizing mutation enabling a subsequent immune escape mutation has also been observed in the evolution of other viral proteins [35].

## Broadened pathways of escape from an antibody used for therapeutic treatment of COVID-19

Last, we determined how mutations in each RBD background facilitate escape from binding by LY-CoV1404. LY-CoV1404 is the antibody in bebtelovimab [36], one of the only clinically approved antibodies that maintains potent neutralization of BA.1 and BA.2. We used deep mutational scanning to identify all single mutations in the Wuhan-Hu-1, BA.1, or BA.2 RBDs that confer 10-fold or greater escape from LY-CoV1404 binding (Figs 4, S8 and S3 Data). Consistent with prior studies [34,36], we find that escape from LY-CoV1404 binding in the Wuhan-Hu-1 background is primarily attributable to mutations at residues K444 and V445, along with mutations to residue P499 that tend to be less well tolerated for ACE2 binding. These same sites of escape are also seen in the Omicron BA.1 and BA.2 escape profiles, but there are also a number of additional mutations and sites of escape from LY-CoV1404 binding in the Omicron backgrounds. These additional sites of escape include mutations at the evolutionarily variable site 446 that are well-tolerated for ACE2 binding. Mutations at sites in the LY-CoV1404 escape profiles such as K444 and V445 have been observed repeatedly in wastewater surveillance [37,38] and in newly emerging Omicron sub-lineages [39], suggesting ongoing surveillance will be important for the continued usage of this antibody. The pattern of escape mutations is consistent with those identified for the closely related S2X324 antibody using replicating VSV/SARS-CoV-2 S chimeras [40]. This broadening of sites of escape from LY-CoV1404 in Omicron is also consistent with a prior study showing that antibodies elicited

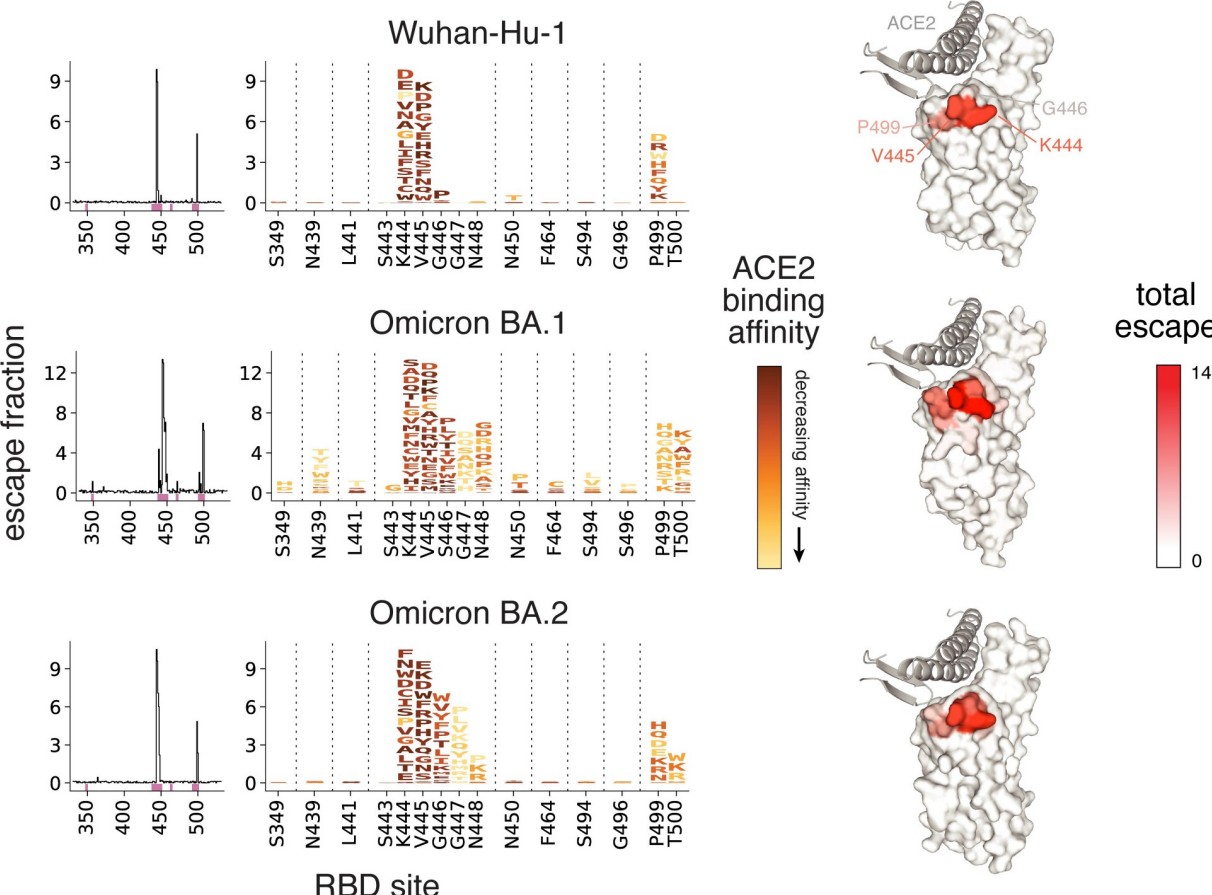

**Fig 4. Complete maps of RBD mutations that escape LY-CoV1404 binding.** Escape from LY-CoV1404 binding in the Wuhan-Hu-1 (top), Omicron BA.1 (middle), and Omicron BA.2 (bottom) backgrounds. The lineplots at left show the total escape of mutations at each site in the RBD. Sites of strong escape indicated by pink bars are shown at the mutation level in logoplots at center. Mutations are colored by their effects on ACE2 binding (scaled according to effects in each background). At right, antibody escape in each background is mapped to the RBD structure bound to ACE2 (key motifs in gray) from PDB 6M0J (Wuhan-Hu-1), 7WPB (Omicron BA.1), and 7XB0 (Omicron BA.2), respectively. "Escape fraction" represents the fraction of cells with that mutation that are sampled in an antibody-escape bin representing 10-fold decreased binding (S8A Fig). See S3 Data for underlying measurement values.

by Wuhan-Hu-1-like viruses are vulnerable to a wider array of escape mutations in the Omicron BA.1 and BA.2 RBDs compared to the ancestral Wuhan-Hu-1 background [41]. We have previously demonstrated that lowered antibody-binding affinities lead to wider escape profiles [42,43], suggesting that LY-CoV1404 may have lower baseline affinity for BA.1 and BA.2 RBDs that opens up these additional pathways of escape.

## Discussion

Omicron emerged suddenly with a constellation of RBD substitutions that balance immune escape with the maintenance of ACE2 binding affinity [10,16,19]. We show that these substitutions also altered the potential future mutational landscape of SARS-CoV-2 evolution in several ways. First, N501Y and perhaps other substitutions induce epistatic shifts at some sites, which may alter which mutations are evolutionarily accessible compared to variants like Delta that lack N501Y. Second, we find that some Omicron substitutions like N501Y and Q498R have become epistatically entrenched, which may disfavor secondary reversions. However, the

substitution Q493R exhibits no entrenchment, consistent with its recent reversion in the evolution of Omicron sub-variants [12]. Last, we found that Omicron substitutions can expand the range of mutations that escape binding from a cross-reactive monoclonal antibody that was elicited by Wuhan-Hu-1-like viruses, potentially increasing the susceptibility of this therapeutic antibody to future viral evolution.

As Omicron continues to evolve, it will explore new regions of protein sequence space. The acquisition of affinity-enhancing mutations in Wuhan-Hu-1 was fundamental to enabling SARS-CoV-2's antigenic evolution by buffering small affinity decreases caused by antibody-escape mutations [10]. Omicron has a different constellation of affinity-enhancing mutations available compared to prior variants, in part because of epistatic shifts and in part because Omicron has already fixed many of the affinity-enhancing mutations available to earlier variants. So far, affinity buffering for subsequent Omicron evolution has mostly involved the R493Q reversion. Whether newly available affinity-enhancing mutations like those at position 453 and 455 will similarly buffer further antigenic evolution—and whether, like N501Y, they epistatically open up other new pathways of sequence evolution [10,20,21]—remains to be seen.

## Materials & methods

### Mutant libraries

We cloned yeast codon-optimized RBD sequences (amino acids N331 –T531 by Wuhan-Hu-1 reference numbering) from Omicron BA.1 and BA.2 into a yeast surface display plasmid. Parental plasmids and associated sequence maps are available from Addgene (accession # 184408 and 184409; original Wuhan-Hu-1 plasmid available as accession # 166782).

Site-saturation mutagenesis libraries spanning all 201 positions in the BA.1 and BA.2 RBDs were produced by Twist Bioscience. We programmed the introduction of precise codon mutations to encode the 19 possible amino acid mutations at each RBD position. To ensure an adequate level of relevant control variants in the library, stop codon mutations were programmed to be introduced at every other position for the first 81 positions, and wildtype codons were re-specified at every other position for the first 102 positions. Libraries were delivered as dsDNA oligonucleotides with constant flanking sequences. The "mutant RBD fragment" sequence delivered for BA.1 as an example (where uppercase letters denote mutated region) is:

tctgcaggctagtggtggaggaggctctggtggaggcggccgcggaggcggagggtcggctagccatatgAACATAACAAACTTATGCCCCTTTGACGAAGTATTTAATGCTACTAGATTCGCATCGGTTTATGCCTGGAATAGAAAGAGGATCAGTAACTGCGTTGCTGATTATTCTGTTTTGTATAACCTGGCCCCTTTCTTCACATTTAAGTGCTACGGGGTCTCGCCTACCAAATTAAACGATTTATGCTTCACCAATGTGTACGCCGATTCTTTTGTGATCAGGGGTGACGAAGTTAGACAGATCGCTCCAGGGCAAACTGGTAATATTGCCGATTACAACTACAAGCTTCCAGACGACTTCACTGGTTGCGTAATAGCATGGAACTCAAACAAGTTAGACTCAAAGGTCTCAGGAAATTATAACTACCTGTACCGTCTTTTCAGGAAATCAAATTTGAAGCCGTTCGAAAGGGACATCTCCACGGAGATATATCAGGCCGGTAACAAGCCCTGCAATGGCGTCGCCGGCTTCAACTGTTACTTCCCCCTAAGGTCATACTCTTTCAGGCCTACATACGGAGTTGGCCATCAGCCATACAGAGTTGTGGTTTTATCTTTCGAGTTGTTGCACGCCCCTGCTACGGTTTGTGGTCCTAAGAAGTCCACTctcgagggggggcggttccgaacaaaagcttatttctgaagaggacttgtaatagagatctgataacaacagtgtagatgtaacaaaatcgactttgttcccactgtacttttagctcg

A second dsDNA fragment encoding constant flanks and a randomized N16 barcode was produced via PCR off of the parental vector with primer-based sequence additions (primers described in [1,44]). This "barcode fragment" sequence is:

cgactttgttcccactgtactttagctcgtacaaaatacaatatacttttcatttctccgtaaacaacatgtttctcccatgtaa-

tatccttttctatttttcgttccgttaccaactttacacatactttatatagctattcacttctatacactaaaaaactaagacaattt-
taattttgctgcctgccatatttcaatttgttataaattcctataatttatcctattagtagctaaaaaaagatgaatgtgaatcgaatcctaa-
gagaattaatgatacggcgaccaccgagatctacactctttccctacacgacgctcttcc-
gatctNNNNNNNNNNNNNNNNNgcggccgcgagctccaattcgccctatagtgagtcgtattacaattcactgg

The "mutant RBD fragment" and "barcode fragment" were combined with NotI/SacI-digested parental plasmid backbone via HiFi Assembly (S1A Fig). An example of the structure of the final assembled library (in the BA.1 background) is available on GitHub: https://github.com/jbloomlab/SARS-CoV-2-RBD_DMS_Omicron/blob/main/data/3294lib_pETcon-SARS2-RBD_Omicron-BA1_lib-assembled.gb. Assembled library plasmids were electroporated into *E. coli* (NEB 10-beta, New England Biolabs C3020K), and plated at limiting dilutions on LB+ampicillin plates. For each library, duplicate plates corresponding to an estimated bottleneck of ~85,000 cfu were scraped and plasmid purified. Plasmid libraries are available from Addgene (accession # 1000000187 and 1000000188). Plasmid libraries were transformed into the AWY101 yeast strain [45] at 10-µg scale according to the protocol of Gietz and Schiestl [46], and aliquots of 18 OD of yeast outgrowth were flash frozen and stored at -80˚C.

As described previously [1,10,44], we sequenced NotI-digested plasmid libraries on a PacBio Sequel IIe to generate long sequence reads spanning the N16 barcode and mutant RBD coding sequence. The resulting circular consensus sequence (CCS) reads are available on the NCBI Sequence Read Archive (SRA), BioProject PRJNA770094, BioSample SAMN30603816. PacBio CCSs were processed using alignparse version 0.2.4 [47] to call N16 barcode sequence and RBD variant genotype and filter for high-quality sequences. Analysis of the PacBio sequencing indicates that all of the intended 3819 RBD mutations were sampled on >1 barcode in the BA.1 libraries, while 19 mutations were sampled 0 or 1 times in the BA.2 library due to failed synthesis of mutations at position 392 (S1E Fig). In contrast to our previous library cloning approach where we added N16 barcodes directly to mutant pool oligos via PCR addition as described in [44], the three-fragment Gibson assembly (S1A Fig) produced more even coverage of specific single mutants and fewer wildtype and double-mutant sequences as intended (S1B–S1D Fig). Complete computational pipelines and summary plots for PacBio data processing and library analysis are available on GitHub: https://github.com/jbloomlab/SARS-CoV-2-RBD_DMS_Omicron/blob/main/results/summary/process_ccs_BA1.md and https://github.com/jbloomlab/SARS-CoV-2-RBD_DMS_Omicron/blob/main/results/summary/process_ccs_BA2.md. Final barcode-variant lookup tables are available on GitHub: https://github.com/jbloomlab/SARS-CoV-2-RBD_DMS_Omicron/tree/main/results/variants

## Deep mutational scanning for ACE2-binding affinity

The effects of mutations on ACE2 binding affinity were determined via FACS-seq assays as previously described [1] with modifications as described in [10]. Titrations were performed in duplicate with pooled mutant libraries of Omicron BA.1 and BA.2 along with the Wuhan-Hu-1 libraries constructed in [10]. Frozen yeast libraries were thawed, grown overnight at 30˚C in SD-CAA media (6.7 g/L Yeast Nitrogen Base, 5.0 g/L Casamino acids, 2.13 g/L MES, and 2% w/v dextrose), and backdiluted to 0.67 OD600 in SG-CAA+0.1%D (SD-CAA with 2% galactose and 0.1% dextrose in place of the 2% dextrose) to induce RBD expression, which proceeded for 16–18 hours at room temperature with mild agitation.

Induced cells were washed with PBS-BSA (BSA 0.2 mg/L), split into 16-OD aliquots, and incubated with biotinylated monomeric human ACE2 protein (ACROBiosystems AC2-H82E8) across a concentration range from $10^{-6}$ to $10^{-13}$ M at 1-log intervals, plus a 0 M sample. Incubations equilibrated overnight at room temperature with gentle mixing. Yeast were washed twice with ice-cold PBS-BSA and fluorescently labeled for 1 hr at 4˚C with 1:100

FITC-conjugated chicken anti-Myc (Immunology Consultants CMYC-45F) to detect yeast-displayed RBD protein and 1:200 PE-conjugated streptavidin (Thermo Fisher S866) to detect bound ACE2. Cells were washed and resuspended in 1x PBS for flow cytometry.

At each ACE2 sample concentration, single RBD$^+$ cells were partitioned into bins of ACE2 binding (PE fluorescence) as shown in S2A Fig using a BD FACSAria II. A minimum of 12.5 million cells were collected at each sample concentration. Collected cells in each bin were grown overnight in 1 mL SD-CAA + pen-strep, and plasmid was isolated using a 96-well yeast miniprep kit (Zymo D2005) according to kit instructions, with the addition of an extended (>2 hr) Zymolyase treatment and a -80˚C freeze/thaw prior to cell lysis. N16 barcodes in each post-sort sample were PCR amplified as described in [1] and submitted for Illumina NextSeq P2 sequencing. Barcode reads are available on the NCBI SRA, BioProject PRJNA770094, Bio-Sample SAMN30603946.

Demultiplexed Illumina barcode reads were matched to library barcodes in barcode-mutant lookup tables using dms_variants (version 0.8.9), yielding a table of counts of each barcode in each FACS bin, available at https://github.com/jbloomlab/SARS-CoV-2-RBD_DMS_Omicron/blob/main/results/counts/variant_counts.csv. Read counts in each FACS bin were downweighted by the ratio of total sequence reads from a bin to the number of cells that were sorted into that bin from the FACS log.

We estimated the level of ACE2 binding of each barcoded mutant at each ACE2 concentration based on its distribution of counts across FACS bins as the simple mean bin [1]. We determined the ACE2-binding constant $K_D$ for each barcoded mutant via nonlinear least-squares regression using the standard non-cooperative Hill equation relating the mean sort bin to the ACE2 labeling concentration and free parameters $a$ (titration response range) and $b$ (titration curve baseline):

$$\text{bin} = a \times [\text{ACE2}]/([\text{ACE2}] + \text{K}_D) + b$$

The measured mean bin value for a barcode at a given ACE2 concentration was excluded from curve fitting if fewer than 2 counts were observed across the four FACS bins or if counts exhibited bimodality (>40% of counts of a barcode were found in each of two non-consecutive bins). To avoid errant fits, we constrained the value $b$ to (1, 1.5), $a$ to (2, 3), and $K_D$ to ($10^{-15}$, $10^{-5}$). The fit for a barcoded variant was discarded if the average cell count across all sample concentrations was below 2, or if more than one sample concentration was missing. We also discarded curve fits where the normalized mean square residual (residuals normalized relative to the fit response parameter $a$) was >40 times the median value across all titration fits. Final binding constants were expressed as -$\log_{10}(K_D)$, where higher values indicate higher binding affinity. The complete computational pipeline for calculating and filtering per-barcode binding constants is available at https://github.com/jbloomlab/SARS-CoV-2-RBD_DMS_Omicron/blob/main/results/summary/compute_binding_Kd.md, and per-barcode affinity values are available at https://github.com/jbloomlab/SARS-CoV-2-RBD_DMS_Omicron/blob/main/results/binding_Kd/bc_binding.csv.

The affinity measurements of replicate barcodes representing an identical amino acid mutant were averaged within each experimental duplicate. The correlations in collapsed affinities in each duplicate experiment are shown in S2B Fig. The final measurement was determined as the average of duplicate measurements. The median BA.1 and BA.2 mutant's final ACE2 affinity measurement collapsed across 43 total replicate barcodes. The final -$\log_{10}(K_D)$ for each mutant and number of replicate barcode collapsed into this final measurement for each RBD mutant are given in S1 Data and https://github.com/jbloomlab/SARS-CoV-2-RBD_DMS_Omicron/blob/main/results/final_variant_scores/final_variant_scores.csv.

## RBD expression deep mutational scanning

Pooled libraries were grown and induced for RBD expression as described above. Induced cells were washed and labeled with 1:100 FITC-conjugated chicken anti-Myc to label for RBD expression via a C-terminal Myc tag, and washed in preparation for FACS. Single cells were partitioned into bins of RBD expression (FITC fluorescence) using a BD FACSAria II as shown in S3A Fig. A total of >17 million viable cells (estimated by plating dilutions of post-sort samples) were collected across bins for each library. Cells in each bin were grown out in SD-CAA + pen-strep, plasmid isolated, and N16 barcodes sequenced as described above. Barcode reads are available on the NCBI SRA, BioProject PRJNA770094, BioSample SAMN30603946.

Demultiplexed Illumina barcode reads were matched to library barcodes in barcode-mutant lookup tables using dms_variants (version 0.8.9), yielding a table of counts of each barcode in each FACS bin, available at https://github.com/jbloomlab/SARS-CoV-2-RBD_DMS_Omicron/blob/main/results/counts/variant_counts.csv. Read counts in each bin were down-weighted using the post-sort colony counts instead of the FACS log counts as with ACE2 titrations above to account for unequal viability of cells in FITC fluorescence bins (i.e., many cells in bin 1 are non-expressing because they have lost the low-copy expression plasmid and do not grow out post-FACS in selective media).

We estimated the level of RBD expression (log-mean fluorescence intensity, logMFI) of each barcoded mutant based on its distribution of counts across FACS bins and the known log-transformed fluorescence boundaries of each sort bin using a maximum likelihood approach [1,48] implemented via the fitdistrplus package in R [49]. Expression measurements were discarded for barcodes for which fewer than 10 counts were observed across the four FACS bins. The full pipeline for computing per-barcode expression values is available at https://github.com/jbloomlab/SARS-CoV-2-RBD_DMS_Omicron/blob/main/results/summary/compute_expression_meanF.md. Final mutant expression values were collapsed within and across replicates as described above, with correlation between experimental replicates shown in S3B Fig. A median of 51 barcodes were collapsed into final BA.1 and BA.2 mutant expression measurements. Final mutant expression values and number of replicate barcode collapsed into this final measurement for each RBD mutant are available in S2 Data and https://github.com/jbloomlab/SARS-CoV-2-RBD_DMS_Omicron/blob/main/results/final_variant_scores/final_variant_scores.csv.

## Quantification of epistasis

Epistatic shifts at each site between pairs of RBD variants were quantified exactly as described by Starr et al. [10]. Briefly, affinity phenotypes of each mutant at a site were transformed to a probability analog via a Boltzmann weighting, and the "epistatic shift" metric was calculated as the Jensen-Shannon divergence between the vectors of 20 amino acid probabilities. The Jensen-Shannon divergence ranges from 0 for two vectors of probabilities that are identical to 1 for two vectors that are completely dissimilar. To avoid noisier measurements artifactually inflating the epistatic shift metric, a given amino acid mutation was only included in the computation if it was sampled with a minimum of 3 replicate barcodes in each RBD background. The calculation of epistatic shifts can be found at: https://github.com/jbloomlab/SARS-CoV-2-RBD_DMS_Omicron/blob/main/results/summary/epistatic_shifts.md.

## LY-CoV1404 antibody escape deep mutational scanning

LY-CoV1404 antibody variable domain sequences were acquired from the structure reported in [36]. Sequence is available at https://github.com/jbloomlab/SARS-CoV-2-RBD_Omicron_

MAP_LY-CoV1404/blob/main/data/LY-CoV1404.fasta. Recombinant antibody was cloned and produced by Genscript. Briefly, antibody variable domains were cloned with the human IgG1 heavy chain and human IgL2 constant regions, expressed in mammalian cells, and IgG purified over MabSelect PrismA columns.

RBD mutant libraries in the Wuhan-Hu-1, BA.1, or BA.2 background were grown and induced for RBD expression as described above. 5 OD of cells were labeled for one hour at room temperature in 1 mL with a concentration of antibody determined as the EC90 from pilot isogenic binding assays. In parallel, 0.5 OD of respective parental constructs were incubated in 100 μL of antibody at the matched EC90 concentration or 1/10 the concentration for FACS gate-setting. Cells were washed, incubated with 1:100 FITC-conjugated chicken anti-Myc antibody to label RBD expression and 1:200 PE-conjugated goat anti-human-IgG (Jackson ImmunoResearch 109-115-098) to label bound antibody. Labeled cells were washed and resuspended in PBS for FACS.

Antibody-escape cells in each library were selected via FACS on a BD FACSAria II. FACS selection gates were drawn to capture approximately 50% of yeast expressing the parental RBD labeled at 10x reduced antibody labeling concentration (see gates in S8A Fig). For each sample, 3.5–4 million RBD$^+$ cells were processed on the sorter with collection of cells in the antibody-escape bin. Sorted cells were grown overnight in SD-CAA + pen-strep, plasmid purified, and barcodes sequenced as described above. In parallel, plasmid samples were purified from 30 OD of pre-sorted library cultures and sequenced to establish pre-selection barcode frequencies. Barcode reads are available on the NCBI SRA, BioProject PRJNA770094, BioSample SAMN30603977.

Demultiplexed Illumina barcode reads were matched to library barcodes in barcode-mutant lookup tables using dms_variants (version 0.8.9), yielding a table of counts of each barcode in each pre- and post-sort population which is available at https://github.com/jbloomlab/SARS-CoV-2-RBD_Omicron_MAP_LY-CoV1404/tree/main/results/counts.

The escape fraction of each barcoded variant was computed from sequencing counts in the pre-sort and antibody-escape populations via the formula:

$$E_v = F \times \left( n_v^{post}/N_{post} \right) \div \left( n_v^{pre}/N_{pre} \right)$$

where $F$ is the total fraction of the library that escapes antibody binding (numbers in S8A Fig), $n_v$ is the counts of variant $v$ in the pre- or post-sort samples with a pseudocount addition of 0.5, and $N$ is the total sequencing count across all variants pre- and post-sort. These escape fractions represent the estimated fraction of cells expressing a particular variant that fall in the escape bin. Per-barcode escape scores are available at https://github.com/jbloomlab/SARS-CoV-2-RBD_Omicron_MAP_LY-CoV1404/tree/main/results/escape_scores.

We applied computational filters to remove mutants with low pre-selection sequencing counts or highly deleterious mutations that escape antibody binding due to e.g. poor RBD surface expression. Specifically, we removed variants that had ACE2 binding scores $< -2$ or expression scores of $< -1.25$, $< -0.834$, and $< -0.954$ in the Wuhan-Hu-1, BA.1, and BA.2 backgrounds respectively (reflecting the lowered parental expression phenotypes of the BA.1 and BA.2 backgrounds), and we removed mutations to the conserved RBD cysteine residues. There were 2,482, 1,850, and 2,024 mutations in the Wuhan-Hu-1, BA.1, and BA.2 backgrounds that passed these computational filters.

Per-mutant escape fractions were computed as the average across barcodes within replicates, with the correlations between replicate library selections shown in S8B and S8C Fig. Final escape fraction measurements averaged across replicates are given in S3 Data and are

available at https://github.com/jbloomlab/SARS-CoV-2-RBD_Omicron_MAP_LY-CoV1404/tree/main/results/supp_data

## Plasmid construction and recombinant protein production

The SARS-CoV-2 Wuhan-Hu-1 RBD construct was synthesized by GenScript into pcDNA3.1-with an N-terminal mu-phosphatase signal peptide and a C-terminal octa-histidine tag, flexible linker, and avi tag (GHHHHHHHHGGSSGLNDIFEAQKIEWHE). The boundaries of the construct are N-328RFPN331 and 528KKST531-C. The SARS-CoV-2 BA.1 RBD construct (G339D, S371L, S373P, S375F, K417N, N440K, G446S, S477N, T478K, E484A, Q493R, G496S, Q498R, N501Y, Y505H) and BA.2 construct (G339D, S371F, S373P, S375F, T376A, D405N, R408S, K417N, N440K, S477N, T478K, E484A, Q493R, Q498R, N501Y, Y505H) were synthesized by Genscript into pcDNA3.1(+) with a BM40+A (for Kozak) signal peptide with a C-terminal avi-tag, flexible linker, and octa-histidine tag (GGLNDIFEAQKIEWHEGSGHHHHHHHH*). The boundaries of the construct are N-328RFPN331 and 528KKST531-C. Genscript mutagenized each construct to produce the single mutational variants SARS-CoV-2 Wuhan-Hu-1+Q493R, BA.1+R493Q, and BA.2+R493Q.

SARS-CoV-2 RBDs were expressed in Expi293F Cells (ThermoFisher Scientific) grown in suspension using Expi293 Expression Medium (ThermoFisher Scientific) at 37°C in a humidified 8% $CO_2$ incubator rotating at 130 RPM. Cells grown to a density of 3 million cells per milliliter were transfected using the ExpiFectamine 293 Transfection Kit (ThermoFisher Scientific) and grown for 4 days. RBDs were purified from clarified supernatants using a nickel HisTrap HP affinity column (Cytiva) and washed with ten column volumes of a 25 mM sodium phosphate 300 mM NaCl pH 8.0 buffer before elution on a gradient to 500 mM imidazole, 25 mM sodium phosphate, 300 mM NaCl pH 8.0 and buffer exchanged using centrifugal filters (Amicon Ultra). Proteins were biotinylated overnight at 4°C using a BirA biotin-protein ligase reaction kit (Avidity) and re-purified using the same HisTrap HP affinity method described above before being flash frozen.

## Biolayer interferometry

Each reagent was prepared in 10x Octet Kinetics Buffer (Sartorius). Biotinylated His-avi-tagged SARS-CoV 2-Wuhan-Hu-1, BA.1, BA.2, Wuhan-Hu-1+Q493R, BA.1+R493Q, and BA.2+R493Q RBDs were diluted to 5 µg/mL and immobilized onto pre-hydrated streptavidin biosensors (Sartorius) to a level of 1 nm total shift. The loaded sensors were then dipped into a 1:3 serial dilution of his-tagged monomeric human ACE2 for 300 seconds followed by dissociation for 300 seconds in 10X kinetics buffer. All measurements were taken with an OctetRED96e (FortéBio) and were performed at 30°C with a shake speed of 1000 RPM. The data were all baseline subtracted and the curves fitted with Octet Data Analysis HT software (v11.1.3.50) and plotted with GraphPad Prism (9.4.1).

## Supporting information

**S1 Fig. Mutant library generation and statistics. (A)** Scheme for generation of the Omicron BA.1 and BA.2 RBD mutant libraries. Site saturation mutagenesis oligonucleotide libraries were constructed by Twist Bioscience with constant flank sequences. For each Omicron background, a three-fragment Gibson Assembly was performed with: (1) the pooled mutant RBD oligonucleotide, (2) a PCR-generated oligonucleotide encoding a randomized N16 nucleotide barcode, and (3) linearized vector backbone. PacBio sequencing of the barcoded mutant library plasmid was used to link N16 barcode to mutant RBD sequence, enabling complete definition of library statistics and the creation of a barcode-variant lookup table such that

subsequent deep mutational scans only require N16 barcode sequencing. **(B-E)** For pooled duplicate Omicron BA.1 (top) and BA.2 (bottom) libraries, (B) the average number of mutations of each class per barcoded variant, (C) the distribution of the number of amino acid mutations per barcoded variant, (D) the mutation rate at each site along the RBD sequence, and (E) the distribution of the total number of associated N16 barcodes for each possible amino acid mutation (from filtered ACE2 binding scores).
(TIF)

**S2 Fig. Deep mutational scanning measurements of mutational impacts on ACE2 receptor-binding affinity. (A)** Representative FACS scheme (replicate 1) used for ACE2-binding deep mutational scanning titration assays on pooled Wuhan-Hu-1, Omicron BA.1, and Omicron BA.2 mutant libraries. Bins of PE fluorescence (ACE2 binding) were drawn on cells pre-selected on FSC/SSC and FITC(RBD)/FSC plots to isolate single RBD$^+$ cells. At each ACE2 concentration, 12.5 million cells total were collected across the four bins. Post-sort cells were sequenced to identify the distribution of counts of each library variant at each concentration, which were fit to titration curves to determine per-variant dissociation constants ($K_D$). **(B)** Correlation in mutant ACE2-binding affinities in the pooled Wuhan-Hu-1, Omicron BA.1, and Omicron BA.2 libraries in experimental duplicates (independently barcoded and assayed mutant libraries). Red dashed line represents the 1:1 linear line. **(C)** Relationship between mutant ACE2-binding affinities in each of the RBD backgrounds that were assayed in parallel. Red dashed line represents the 1:1 linear line. Plots illustrate similar effects of mutations in the Omicron BA.1 and BA.2 backgrounds compared to the substantial variation in mutational effects in either Omicron background compared to Wuhan-Hu-1.
(TIF)

**S3 Fig. Deep mutational scanning measurements of mutational impacts on folded RBD expression. (A)** FACS scheme used for RBD expression deep mutational scanning assays on pooled Wuhan-Hu-1, Omicron BA.1, and Omicron BA.2 mutant libraries. Bins of FITC fluorescence (RBD expression) were drawn on cells pre-selected on FSC/SSC plots to isolate single cells. Approximately 17 million viable cells were collected across the four bins and sequenced to identify the distribution of each library variant across bins, enabling the back-calculation of per-variant FITC mean fluorescence intensity (MFI). **(B)** Correlation in mutant RBD expression values in the pooled Wuhan-Hu-1, Omicron BA.1, and Omicron BA.2 libraries in experimental duplicates (independently barcoded and assayed mutant libraries). Red dashed line represents the 1:1 linear line. **(C)** RBD yeast-surface expression levels of Wuhan-Hu-1, Omicron BA.1, and Omicron BA.2 as measured by flow cytometry **(D)** RBD yeast-surface expression levels of Wuhan-Hu-1, Omicron BA.1, and Omicron BA.1 with the "rpk9" stabilizing mutations [27] as measured by flow cytometry.
(TIF)

**S4 Fig. Epistatic shifts in all variant backgrounds assessed to date.** Faceted lineplots illustrate epistatic shifts in the effects of mutations on ACE2 binding at each RBD position compared to Wuhan-Hu-1 in all variant backgrounds for which we've conducted deep mutational scans. Interactive plot is available at https://jbloomlab.github.io/SARS-CoV-2-RBD_DMS_Omicron/epistatic-shifts/. All measurements are compared to the Wuhan-Hu-1 dataset generated in this study. Wuhan-Hu-1 (replicate) corresponds to our prior Wuhan-Hu-1 dataset that was gathered in parallel with the Alpha, Beta, and Eta datasets that are also represented here [10].
(TIF)

**S5 Fig. Mutation-level epistatic shifts in Omicron BA.1 and BA.2 RBDs compared to Wuhan-Hu-1.** (A, B) Mutation-level plots of epistatic shifts at sites of interest. Each scatter plot shows the measured affinity of the 20 amino acids in the Omicron BA.1 (A) or BA.2 (B) versus Wuhan-Hu-1 RBDs. Red dashed lines indicate parental affinities. Orange letters are mutations that were sampled with fewer than 3 unique barcodes across titration replicates in one or both backgrounds and were not included in the epistatic shift computation. (TIF)

**S6 Fig. Patterns of entrenchment of all Omicron BA.1 and BA.2 sequence substitutions.** (A, B) For each substitution distinguishing Wuhan-Hu-1 from Omicron BA.1 (A) or BA.2 (B), the pattern of epistasis is reflected in the effect of the forward mutation in the Wuhan-Hu-1 background (WH1, black) versus its reversion in the Omicron (orange or pink) background. Entrenchment is evident in mutations where the slope of the orange or pink line is greater (more positive or less negative) than the slope of the black line (e.g., S371L, N501Y), and arises from favorable epistasis between the entrenched substitution and one or more other co-occurring substitutions. Mutations where the slopes are the same do not show epistasis with the other substituted Omicron positions (e.g., K417N). (TIF)

**S7 Fig. Biolayer interferometry measurements of binding kinetics. (A, B)** BLI sensorgrams from experiments with RBDs immobilized on a BLI biosensor with monomeric human ACE2 as analyte. Binding curves were measured at ACE2 concentrations from 167 (red) to 0.7 (pink) nM at 3-fold dilutions, and kinetic parameters were inferred from a global fit. Curves in (A) represent the curves underlying the kinetic values given in Fig 3B, and curves in (B) represent a second replicate on independently purified RBD batches. (No second replicate for Omicron BA.2 was measured.) **(C)** Kinetic values for binding measurements shown in (B). Asterisk on BA.2 values indicates that these values are repeated from the replicate 1 experiments (A and Fig 3B). (TIF)

**S8 Fig. Deep mutational scanning of mutations that escape LY-CoV1404 antibody binding. (A)** FACS gates used to identify mutations that escape binding by LY-CoV1404 in each RBD background. For each experiment, an antibody-escape gate was drawn that captures approximately 50% of cells in the respective wildtype control labeled at 10% of the library selection antibody concentration. The "escape fraction" represents the fraction of cells of a mutant genotype that fall into the antibody-escape sort bin. **(B, C)** For each background, the correlation in the per-mutation escape fraction (B) or the sum of escape fractions of all mutations at a site (C). (TIF)

**S1 Data. The effects of all single amino acid mutations in the Wuhan-Hu-1, Omicron BA.1, and Omicron BA.2 RBD on ACE2-binding affinity.** These data are also available at: https://github.com/jbloomlab/SARS-CoV-2-RBD_DMS_Omicron/blob/main/results/final_ variant_scores/final_variant_scores.csv (CSV)

**S2 Data. The effects of all single amino acid mutations in the Wuhan-Hu-1, Omicron BA.1, and Omicron BA.2 RBD on RBD expression.** These data are also available at: https:// github.com/jbloomlab/SARS-CoV-2-RBD_DMS_Omicron/blob/main/results/final_variant_ scores/final_variant_scores.csv (CSV)

**S3 Data. The effects of all single amino acid mutations in the Wuhan-Hu-1, Omicron BA.1, and Omicron BA.2 RBD on escape from LY-CoV1404 antibody binding.** These data are also available at: https://github.com/jbloomlab/SARS-CoV-2-RBD_Omicron_MAP_LY-CoV1404/tree/main/results/supp_data
(CSV)

## Acknowledgments

We thank the Fred Hutchinson Cancer Center Flow Cytometry and Genomics core facilities for experimental support, and the Fred Hutchinson Scientific Computer group supported by ORIP grant S10OD028685.

## Author Contributions

**Conceptualization:** Tyler N. Starr, Allison J. Greaney, Jesse D. Bloom.

**Formal analysis:** Tyler N. Starr.

**Funding acquisition:** David Veesler, Jesse D. Bloom.

**Investigation:** Tyler N. Starr, Allison J. Greaney, Cameron M. Stewart, Alexandra C. Walls, David Veesler.

**Methodology:** Tyler N. Starr, Allison J. Greaney.

**Software:** William W. Hannon.

**Supervision:** David Veesler, Jesse D. Bloom.

**Visualization:** Tyler N. Starr.

**Writing – original draft:** Tyler N. Starr, Jesse D. Bloom.

**Writing – review & editing:** Tyler N. Starr, Allison J. Greaney, Cameron M. Stewart, Alexandra C. Walls, William W. Hannon, David Veesler, Jesse D. Bloom.

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
