## [Decision Letter · Decision Letter 0]

12 Oct 2022

Dear Dr Bloom,

Thank you very much for submitting your manuscript "Deep mutational scans for ACE2 binding, RBD expression, and antibody escape in the SARS-CoV-2 Omicron BA.1 and BA.2 receptor-binding domains" for consideration at PLOS Pathogens. As with all papers reviewed by the journal, your manuscript was reviewed by members of the editorial board and by several independent reviewers. In light of the reviews (below this email), we would like to invite the resubmission of a significantly-revised version that takes into account the reviewers' comments. 

As you will see, the comments made by reviewers request editorial changes, additional clarification or some additional statistical considerations; addressing these comments will likely improve the paper without significantly altering its structure.

We cannot make any decision about publication until we have seen the revised manuscript and your response to the reviewers' comments. This will likely not require an additional round of revisions.

Sincerely,

Chris Ka Pun Mok

Guest Editor

PLOS Pathogens

Marco Vignuzzi

Section Editor

PLOS Pathogens

Kasturi Haldar

Editor-in-Chief

PLOS Pathogens

orcid.org/0000-0001-5065-158X

Michael Malim

Editor-in-Chief

PLOS Pathogens

orcid.org/0000-0002-7699-2064

Reviewer's Responses to Questions

**Part I - Summary**

Reviewer #1: Starr et al. carried out deep mutational scans on SARS-CoV-2 Omicron BA.1 and BA.2 S RBD using their previously established workflow, as a continued effort in probing the RBD mutational landscape. They reported that, like previous variants, BA.1/BA. 2 are highly tolerant to mutation and can gain mutations that increase ACE2-binding affinity, yet some of those mutations were reversions to Wuhan-Hu-1. The authors also found reduced yeast surface expression of BA.1/BA.2 RBD compared to Wuhan-Hu-1 and listed potential stabilizing mutations within the Omicron RBD core. Epistatic shifts are also identified, including substitutions that show epistatic entrenchment. At the end, the author screened for LY-CoV1404 escape substitutions.

Overall, the experimental design and data presentation are straightforward and appropriate. While the results on epistatic shifts in this study are conceptually similar to one of the authors’ previous publications (PMID: 35762884), the finding on broadened pathways of escape from LY-CoV1404 is timely and corroborate with a recent preprint by another group (ref 41). The data in this study also provide a useful resource for the SARS-CoV-2 research field. I only have a few minor comments.

Reviewer #2: Previous work by the authors mapped the impact of single amino acid mutations in the receptor-binding domain (RBD) of SARS-CoV-2 on ACE2 binding and antibody escape. However, as the authors note, the impact of amino acid mutations in the RBDs of emerging SARS-CoV-2 variants may vary, due to differences in the genetic background of the novel virus subtype – a phenomenon known as epistasis. Thus, here the authors characterized the functional impact of single amino acid mutations in the RBD of the more recent Omicron BA.1 and BA.2 lineages of SARS-CoV-2 on binding to ACE2, and on escape from binding by a therapeutic monoclonal antibody (LY-CoV1404) using deep mutational scanning. Overall, the study is well-executed and compelling. Consistent with their hypothesis, the authors find the impact of some – but not all – amino acid substitutions on RBD function to differ in the Omicron lineages as compared to the ancestral Wuhan-Hu-1 strain, underscoring the importance of contextualizing predictions of SARS-CoV-2 protein function within the relevant genetic background. The authors further identify mutations in the Omicron BA.1 and BA.2 RBD likely to catalyze viral escape from the only clinically approved monoclonal antibody that potently neutralizes Omicron BA.1 and BA.2, which should inform SARS-CoV-2 genomic surveillance efforts and public health strategy as the Omicron lineages continue to evolve.

Reviewer #3: The authors built variant libraries of SARS-CoV-2 BA.1/BA.2 RBD using yeast surface display and performed deep mutational scanning assays to determine the impact on RBD expression and ACE2-binding affinity of each substition on BA.1/BA.2 RBD. They also analyzed the epistatic effects of Omicron's mutations and the escape profile of an important Omicron-neutralizing therapeutic NAb (LY-CoV1404). The data is trustworthy and helps a lot to explain the evolution of SARS-CoV-2 Omicron variants. These data has been widely accepted and cited by the community even before the submission of the manuscript. Therefore, I recommend this manuscript for publication despite two minor concerns to be addressed.

**Part II – Major Issues: Key Experiments Required for Acceptance**

Reviewer #1: None

Reviewer #2: 1. In several locations in the manuscript (e.g. line 27), the authors claim to have determined the impacts of “all amino acid mutations” in the RBD. This can be considered misleading given that combinatorial mutations were not assessed, therefore it is more precise to say, “all single-site amino acid mutations”, or something similar.

2. Previous work by the authors used dimeric ACE2 (e.g. PMC7418704). What was their rationale for switching to monomeric ACE2 in this study? Do the results from DMS of the Wuhan-Hu-1 RBD remain consistent across the 2 ACE2 screening approaches?

3. When describing Figure 1 in the text, several amino acid sites are alluded to. However, it is challenging to cross reference these sites in the figure with the relatively small heatmap tiles. Could sites of interest be highlighted more clearly in some way in the current figure. Alternatively, perhaps the heatmap could be expanded (and/or rotated 90 degrees) so the tiles are more clearly visible? Also, it is not immediately obvious what the yellow regions on the heatmaps in Figure 1 represent.

4. The authors argue that the N501Y substitution shared between the Omicron and Beta RBDs is likely a “dominant determinant” that underlies the similar pattern of epistatic shifts in the Omicron and Beta variants. However, it is unclear how the authors arrived at this interpretation. A more robust discussion of this reasoning would be useful.

5. The authors suggest throughout the manuscript that the pattern of epistatic shift in the Omicron and Beta variants is similar. This broadly appears to be the case when examining Figure 2. However, this claim could be made stronger in a couple of ways. First, if their interpretation is correct, including another SARS-CoV-2 lineage as a benchmark in this analysis (e.g. Delta) would allow the authors to specifically show that the epistatic shift in mutational effects (vs. Wuhan-Hu-1) of Omicron are more similar to the Beta lineage when compared to other SARS-CoV-2 lineages; currently while Beta and Omicron visually look similar, there is no comparator. However, an additional DMS of the Delta variant RBD may be beyond the score of this paper. Thus, the authors might additionally consider using a quantitative, statistical metric to score similarity (or lack thereof) in epistatic shifts between variants (e.g. Beta and Omicron BA.1) in a pairwise manner to bolster their argument.

Reviewer #3: None

**Part III – Minor Issues: Editorial and Data Presentation Modifications**

Reviewer #1: 1. Lines 104-105: “We found that Omicron BA.1 and BA.2 RBDs have reduced yeast surface expression levels relative to Wuhan-Hu-1 (S2 Data and S3C Fig.)”. The data for BA.2 seem to be missing in Figure S3C?

2. Figure 1C: It seems like the mutation with no data available was labeled in yellow rather than the designated grey.

3. Line 110: Should F492W be F392W instead?

4. Lines 139-143: Is there a structural explanation on the distinctive effects of N439K and L455W on the ACE2 affinity between Wuhan-Hu-1 and Omicron RBDs, given that both ACE2-bound structures are available (PDB 6M0J and 7XO9)?

Reviewer #2: (No Response)

Reviewer #3: 1. Line 229. It's better to mention some designated names of newly emerging variants spreading rapidly that carry K444/V445 mutations and are highly likely to escape LY-CoV1404, such as BJ.1/XBB, BQ.1 and BR.1.

2. Fig. 4. Please use the ACE2-RBD complex structure with BA.1/BA.2 RBD (eg. PDB: 7WPB/7XB0) instead of Wuhan-Hu-1 RBD for projection of BA.1/BA.2-based DMS data.

PLOS authors have the option to publish the peer review history of their article (what does this mean?). If published, this will include your full peer review and any attached files.

Reviewer #1: No

Reviewer #2: No

Reviewer #3: No
---

## [Decision Letter · Decision Letter 1]

26 Oct 2022

Dear Dr Bloom,

We are pleased to inform you that your manuscript 'Deep mutational scans for ACE2 binding, RBD expression, and antibody escape in the SARS-CoV-2 Omicron BA.1 and BA.2 receptor-binding domains' has been provisionally accepted for publication in PLOS Pathogens.

Best regards,

Chris Ka Pun Mok

Guest Editor

PLOS Pathogens

Marco Vignuzzi

Section Editor

PLOS Pathogens

Kasturi Haldar

Editor-in-Chief

PLOS Pathogens

orcid.org/0000-0001-5065-158X

Michael Malim

Editor-in-Chief

PLOS Pathogens

orcid.org/0000-0002-7699-2064

Reviewer Comments (if any, and for reference):

Reviewer's Responses to Questions

**Part I - Summary**

Reviewer #1: The authors have addressed all my previous concerns.

Reviewer #3: All of my concerns have been well-addressed in the revised manuscript. Also, the modification made according to other reviewers' comments have improved the clarity. I support its publication in its current form.

**Part II – Major Issues: Key Experiments Required for Acceptance**

Reviewer #1: (No Response)

Reviewer #3: (No Response)

**Part III – Minor Issues: Editorial and Data Presentation Modifications**

Reviewer #1: (No Response)

Reviewer #3: (No Response)

PLOS authors have the option to publish the peer review history of their article (what does this mean?). If published, this will include your full peer review and any attached files.

Reviewer #1: No

Reviewer #3: No

---

## [Editor Report · Acceptance letter]

31 Oct 2022

Dear Dr Bloom,

We are delighted to inform you that your manuscript, "Deep mutational scans for ACE2 binding, RBD expression, and antibody escape in the SARS-CoV-2 Omicron BA.1 and BA.2 receptor-binding domains," has been formally accepted for publication in PLOS Pathogens.

Best regards,

Kasturi Haldar

Editor-in-Chief

PLOS Pathogens

orcid.org/0000-0001-5065-158X

Michael Malim

Editor-in-Chief

PLOS Pathogens

orcid.org/0000-0002-7699-2064